# Molecular Biomarker Expression in Window of Opportunity Studies for Oestrogen Receptor Positive Breast Cancer—A Systematic Review of the Literature

**DOI:** 10.3390/cancers14205027

**Published:** 2022-10-14

**Authors:** James W. M. Francis, Manmeet Saundh, Ruth M. Parks, Kwok-Leung Cheung

**Affiliations:** 1School of Medicine, University of Nottingham, Nottingham NG7 2UH, UK; 2Nottingham Breast Cancer Research Centre, University of Nottingham, Nottingham NG7 2RD, UK

**Keywords:** breast cancer, window of opportunity, oestrogen receptor-positive

## Abstract

**Simple Summary:**

Window of opportunity (WoO) trials allow the opportunity to assess the use of drugs in breast cancer research before treatment has commenced. The aim of this review of the literature is to review WoO trials in patients with oestrogen receptor-positive (ER+) breast cancer to help guide treatment. This will be useful for patients who receive drug treatment before surgery, or as an alternative to surgery in older, more frail adults.

**Abstract:**

Window of opportunity (WoO) trials create the opportunity to demonstrate pharmacodynamic parameters of a drug in vivo and have increasing use in breast cancer research. Most breast cancer tumours are oestrogen receptor-positive (ER+), leading to the development of multiple treatment options tailored towards this particular tumour subtype. The aim of this literature review is to review WoO trials pertaining to the pharmacodynamic activity of drugs available for use in ER+ breast cancer in order to help guide treatment for patients receiving neoadjuvant and primary endocrine therapy. Five databases (EMBASE, Cochrane, MEDLINE, PubMed, Web of Science) were searched for eligible studies. Studies performed in treatment-naïve patients with histologically confirmed ER+ breast cancer were included if they acquired pre- and post-treatment biopsies, compared measurement of a proteomic biomarker between these two biopsies and delivered treatment for a maximum mean duration of 31 days. Fifteen studies were eligible for inclusion and covered six different drug classes: three endocrine therapies (ETs) including aromatase inhibitors (AIs), selective oestrogen receptor modulators (SERMs), selective oestrogen receptor degraders (SERDs) and three non-ETs including mTOR inhibitors, AKT inhibitors and synthetic oestrogens. Ki67 was the most frequently measured marker, appearing in all studies. Progesterone receptor (PR) and ER were the next most frequently measured markers, appearing five and four studies, respectively. All three of these markers were significantly downregulated in both AIs and SERDs; Ki67 alone was downregulated in SERMs. Less commonly assessed markers including pS6, pGSH3B, FSH and IGF1 were downregulated while CD34, pAKT and SHBG were significantly upregulated. There were no significant changes in the other biomarkers measured such as phosphate and tensin homolog (PTEN), Bax and Bcl-2.WoO studies have been widely utilised within the ER+ breast cancer subtype, demonstrating their worth in pharmacodynamic research. However, research remains focused upon routinely measured biomarkers such ER PR and Ki67, with an array of less common markers sporadically used.

## 1. Introduction

Breast cancer is now the most commonly diagnosed form of cancer worldwide [1]. It is a heterogeneous disease given the variability in tumour receptor expression [2]. The oestrogen-receptor-positive (ER+) subtype is the most prevalent molecular subtype, accounting for approximately 70% of all breast cancers [3]. Multiple classes of endocrine therapy (ET) have consequently been developed to inhibit the proliferative effect of oestrogen [4] in ER+ breast cancer.

A major clinical challenge surrounding ETs is their susceptibility to inducing hormone resistance. This resistance can be de novo or acquired [5] following prolonged exposure. While the precise mechanism remains unknown and is likely to be a combination of different factors, it is thought that reduced ER expression alongside cross-talk with associated co-regulators underpin why ETs lose efficacy in approximately 45–60% of patients [6]. Many new therapies are being developed in an attempt to overcome this hurdle.

Drug development requires clear illustration of a drug’s pharmacodynamic impact. One way of determining such information is through window of opportunity (WoO) studies. WoO studies rely on tumour biopsies taken at two time points, with a pharmacological intervention of interest administered throughout the time elapsed in between (usually only 2–4 weeks) [7]. In the context of breast cancer, WoO studies typically (although not always) take these biopsies from patients awaiting tumour resection [8]. The disparity in histological makeup between biopsies can be used to infer the impact of the treatment given. This elucidation of pharmacodynamic impact is the primary goal of WoO studies (rather than any therapeutic benefit conferred by longer neoadjuvant therapy) and this can help refine therapeutic options prior to subsequent Phase III testing. This data is also potentially useful in a clinical context when guiding more clinically vulnerable patient groups through neoadjuvant therapy whereby optimizing patient selection is paramount.

WoO studies have become widely adopted into breast cancer research since short-term pre-surgical tamoxifen was first shown to reduce Ki67 labelling index (LI) in ER+ breast cancer [9]. Since then, multiple WoO studies have been performed to reflect the propagation of therapeutic options available. Given their expanded uptake, there is increasingly a necessity to collate this new WoO data so that we may reassess the value of WoO studies as a tool for contemporary drug research.

This literature reviews aim to assimilate WoO studies across the range of drugs used in ER+ breast cancer and collate data for changes in biomarker expression throughout WoO studies.

## 2. Materials and Methods

### 2.1. Search Strategy and Study Selection

The review of the literature was conducted on 14 March 2022 across MEDLINE, EMBASE, PubMed, Cochrane Library and Web of Science based on a pre-determined search strategy (see Appendix A). The process was undertaken between two reviewers (JF and MS), with ambiguity resolved by a third reviewer (RP).

Inclusion criteria for participants comprised: treatment-naïve status with histologically confirmed ER+ breast cancer (although no specific threshold for ER expression was dictated); acquisition of both pre-treatment and post-treatment biopsies, with analysis of biomarker expression between; duration of drug therapy up to an average of 31 days; published in English; full text available.

Exclusion criteria included: ER-negative breast cancer; previous exposure to drug therapy amongst participants; study exceeding 31 days of mean duration of drug therapy.

### 2.2. Data Extraction

Relevant variables for extraction included: lead author(s) and year of publication, the relevant drug being investigated in each eligible arm of the trial, duration of therapy, number of participants, threshold for confirming ER+ status, and molecular biomarkers measured. These variables were extracted directly from the study text.

Given the heterogeneity in the manner of recording biomarker response, quantitative meta-analysis was not suitable.

## 3. Results

### 3.1. General Characteristics

The literature search identified 2697 potentially relevant studies (1644 after duplicate removal). Overall, 1629 studies were deemed ineligible, leaving 15 to be included in the review. The full process is outlined in Figure 1. A summary of the characteristics of the included studies in given in Table 1.

For the remainder of this article, the studies pertaining to this review will be referred to by their study number as per Table 1.

There was significant heterogeneity across the threshold set for determining ER+ status (ranging from ≥1% to ≥20%), and even in the methodology used (for example, Allred Score vs. raw percentage expression).

Out of the eligible studies, six (40.0%) were single-centre, while the remaining nine studies (60.0%) were multi-centre. A total of 5189 patients were included in the summative analysis; study 15 contributed 86.3% of these.

Some of the studies contained more than one eligible treatment arms; there were 22 study arms included from the 15 studies. All but one (study #8) randomised between their different groups.

### 3.2. Changes in Molecular Biomarker Expression in the Window of Opportunity Setting

Across all 15 studies, the most commonly measured biomarker was Ki67, which was measured in all studies. PR was measured in five studies, while ER was measured in four, caspase-3 appeared in two. Sex hormone binding globulin (SHBG), phosphorylated-S6 protein (pS6), CD34, phospho(p)GSK3B, pPRAS40, pAKT, phosphatase and tensin homolog (PTNENcaspase-3, BAX, B-cell lymphoma (BCL)2, insulin-like growth factor (IGF)1, caspase-3, prolactin and follicle stimulating hormone (FSH) were each measured in one study.

Six different drug classes were utilised across the studies, three endocrine therapies (ETs) including aromatase inhibitors (AIs), selective oestrogen receptor modulators (SERMs), selective oestrogen receptor degraders (SERDs) and three non-ETs including mammalian target of rapamycin (mTOR) inhibitors, AKT inhibitors and synthetic oestrogens. The findings relating to biomarker changes within each of these drug classes will now be presented and are summarised in Table 2.

#### 3.2.1. Aromatase Inhibitors (AIs)

Aromatase inhibitors were investigated within eight treatment arms. They were shown to cause substantial lowering in Ki67 expression in all but one study (study #8). Values for the mean decrease in Ki67 expression ranged from 29.1 (study #7)—85% (study#9) [17].

Study #8 found that 54% of participants recorded a drop in Ki67 expression following AI therapy, however when the data was pooled it was found that there was no significant change between Ki67 levels; the median change was 0%.

ER expression also decreased by 13% following anastrozole and PR expression by 37% in study #9. Caspase-3 (a marker of apoptosis) was shown to decrease by 10.9% and 13.9% in studies #1 and #9, respectively. Upregulation of CD34 protein (a transmembrane glycoprotein) was found in study #8 following anastrozole treatment, however to no greater extent than observed in the control arm [16].

#### 3.2.2. Selective Oestrogen Receptor Degraders (SERDs)

Three arms investigated SERDs overall. Study #10 contained two SERDs, fulvestrant and AZD9496, a novel SERD with preferential bioavailability to fulvestrant which has undergone phase I testing. Both SERDs significantly decreased Ki67 levels by 75.4% and 39.9%, respectively. ER expression was also examined. AZD9496 elicited a 24.3% reduction in ER expression, while the effect of fulvestrant was 36.3% [18].

Similarly, fulvestrant reduced PR expression on average by 68.7% and AZD9496 by 33.3%.

#### 3.2.3. Selective Oestrogen Receptor Modulators (SERMs)

Tamoxifen was investigated in five study arms and was again shown to reduce Ki67 expression. In the POWERPIINC trial (study #3), median Ki67 expression decreased from 17.0% to 9.0% [11] while study 7# demonstrated mean Ki67 expression (measured via Allred scores) decreasing from 3.61 to 2.56 [15]. ER expression was also shown to have fallen marginally from an Allred score of 7.00 to 6.76 while PR increased from an Allred score of 5.40 to 6.73; however, both results were deemed sub-significant [15].

Study #5 also demonstrated increases in sex hormone-binding globulin (SHBG) of 25.4 and 18.6 ng/mL in non-smokers and smokers, respectively [13].

#### 3.2.4. Mammalian Target of Rapamycin (mTOR) Inhibitor

Two study arms investigated mTOR inhibitors in the form of everolimus. Ki67 decreased by 40.84% on average following everolimus therapy in study #6 [14]. The other molecular parameter tested was phosphorylated-S6 protein (pS6) (an mTOR substrate), which was reduced by an average of 79.9%.

Study #12 assessed Ki67 following everolimus therapy. Out of the 21 matched patient profiles, 13 patients responded to treatment as judged by a significant decrease in Ki67 (95% confidence interval) expression from baseline [20]. The same study also found that seven patients registered a significant fall in cytoplasmic phospho-AKT (pAKT) expression (95% confidence interval); a protein that induces signals interfering with apoptosis [20].

#### 3.2.5. AKT Inhibitor

Study 11 was the only study to investigate Ki67 concentrations following administration of capivasertib (a potent oral AKT inhibitor); with percentage of positive nuclei falling by an absolute value of 9.6% (*p* = 0.031) from baseline [19]. Phospho-GSK-3β (pGSK3β) and proline-rich Akt substrate of 40 kDa (pPRAS40), pAKT and FOXO3a are proteins involved in cell apoptosis and proliferation. pGSK3β decreased by absolute H score (range 0–300) of 55.3 (*p* = 0.006) and pPRAS40 decreased by H score of 83.8 (*p* < 0.0001). The marker pAKT was upregulated following capivasertib therapy, resulting in an H score increase of 81.3 (*p* = 0.005); FOXO3a was similarly upregulated by a H score of 29.6, although this was not statistically significant (*p* = 0.229). The marker pS6 decreased by H score 42.3 (*p* = 0.004).

#### 3.2.6. Oestrogens

Estetrol (E4) was the sole oestrogen tested in this review. Study #14 found no significant change in Ki67 expression with E4 treatment. There was however, a substantial ERα downregulation compared to baseline. Upregulation of ERβ was noted, however this finding was not statistically significant. [22].

E4 did cause a significant decrease in SHBG over the 14-day period (*p* < 0.05) [22]. Interestingly, post-menopausal patients treated with E4 had a significant decrease in follicle stimulating hormone (FSH) after 14 days (*p* < 0.05) [22]. Prolactin was also increased, as was IGF-1 however there was no corresponding upregulation of IGF-1 receptors [22]. There was no significant change in BCL2 or BAX levels.

## 4. Discussion

The principal objective of this study was to collate WoO data regarding the pharmacodynamic effect of different pharmacological interventions used in the ER+ breast cancer setting. To this end, the review confirmed previous analysis of breast cancer as a purported area of relative strength for WoO studies [24] and demonstrated the wealth of data for the ER+ subset specifically. The review also provided significant insight into the pharmacodynamic biomarkers related to common treatment modalities—specifically the widespread uptake of Ki67 as a marker of proliferation.

The variability in the threshold for ER expression should be noted in the context of the most recent ASCO/CAP guidelines which outline the disparity in potential response to ET between true ER+ cancers (≥10% expression) compared to ER low + (≤10%) which often behave in a manner more similar to ER− cancers [25].

### 4.1. Changes in Molecular Biomarker Expression in the Window of Opportunity Setting

The findings from our review clearly showed Ki67 was the most studied biomarker across all 15 trials. This is concordant with the wider landscape of breast cancer research, whereby Ki67 assessment has become particularly prevalent [26]. Since its discovery [27] Ki67 has long been known to inform clinicians about specific tumour characteristics, usually through calculation of labelling index via immunohistochemistry [28]. Whilst the biological function of the protein is still unclear, we know that it is synonymous with cell cycle mitotic divisions and thus a marker of cell proliferation [28].

It has also become used to indicate drug efficacy on a molecular level for such therapies, which aim to decrease proliferation of cancer cells. A study by Cabrera-Galeana et al. [29] used Ki67 expression as a biomarker of molecular response to neoadjuvant chemotherapy in 435 patients and found that 57% (comprised of all different breast cancer subtypes) showed with a decrease in Ki67 expression and was correlated with better survival outcomes.

There was minimal evidence, however, that Ki67 response was linked to the extent of ER positivity (as underlined by the widely divergent thresholds set for determining ER+ status) with studies as low as 1% ER positivity showing response to ETs via downregulation of Ki67. This is in spite of the ASCO/CAP guidance which recommends differentiating ER low-positive tumours as a distinct tumour subtype given the potential for differential response to ETs [25]. It should be noted, however, that the relatively low sample size of a number of these studies in conjunction with the variable manners of quantifying Ki67 downregulation limit the meaningful inferences which could be taken from this.

#### 4.1.1. Aromatase Inhibitors (AIs)

Aromatase inhibitors are a well-established drug class in post-menopausal patients for the treatment of ER+ breast cancer [30]. In this review, AIs were associated with decrease in Ki67, ER and PR expression, which is as expected.

Downregulation of ER following AI was first seen in mouse models whereby letrozole therapy prompted significantly reduced cellular expression of ERβ, when compared to either pre-treatment samples or the control arm [31]. Robertson et al. hypothesised that reduced activation of the ER signalling pathway following AI therapy is the cause of ER downregulation [17]. The resultant fall in PR expression is most likely due to the reliance of PR upon a fully functioning ER pathway as an oestrogen-inducible protein [32].

The decrease in the apoptotic marker caspase-3 is in line with previous literature that showed AIs eliciting a slight reduction in apoptosis [33,34]. Finally, CD34 was utilised as a surrogate marker for angiogenesis in study #8 following anastrozole. While the increases in the anastrozole arm from baseline to surgery was ultimately deemed insignificant, the same observation in the control arm could be explained by the hypothesis that invasive procedures could act as a causative agent for angiogenesis [35].

The newer AI exemestane has more recently received National Institute for Health and Care Excellence (NICE) approval, and as such, it would be beneficial to receive future WoO data on this drug and other novel AIs.

#### 4.1.2. Selective Oestrogen Receptor Degraders (SERDs)

Given the mechanism of action of SERDs, ER expression can be used as a marker of effect. Study #10 found ER downregulation following fulvestrant and this is concordant with the accepted mechanism of action of SERDs [36]. As was the case with AIs, the fall in PR expression can largely be attributed to the fall in ER expression due to the aforementioned dependence on the ER signalling pathway.

In regard to the anti-proliferative action of SERD therapy, both fulvestrant and AZD9496 significantly decreased Ki67 expression as expected. This is in keeping with the data from the coopERA trial which found the downregulatory effects of girdestrant upon Ki67 was superior to the AI anastrozole; reinforcing the utility of SERDs in this context [37].

#### 4.1.3. Selective Oestrogen Receptor Modulators (SERMs)

As with SERDs, SERMs ultimately prevent the proliferative effects of oestrogen and consequently aim to limit cellular proliferation. This is evident by significant decreases in Ki67 expression demonstrated in the POWERPIINC study.

Since SERMs generally bind reversibly to ER depending on tissue location, SHBG is becoming increasingly prevalent in the context of tamoxifen research as a measure of peripheral receptor oestrogenicity [38]. The elevated levels of SHBG recorded after tamoxifen therapy were in line with previous studies [39]. This confirmation of induced peripheral ER agonistic activity further demonstrates the dual polarity of SERMs and may explain the elevated risk of endometrial cancer associated with tamoxifen [40,41].

#### 4.1.4. mTOR Inhibitors

Everolimus was the only mTOR inhibitor used in any study. In the UK, everolimus is NICE approved as a second-line therapy (in combination with exemestane) for post-menopausal ER+/HER2− breast cancer following prior treatment with an AI [42].

Our review demonstrated that everolimus effectively caused dysregulation of the PI3K, AKT, mTOR pathway, with pAKT and pS6 being significantly decreased, demonstrating the ability of the drug to bring about apoptotic change and promote cell cycle arrest

#### 4.1.5. AKT Inhibitor

Given the significant crossover between the mTOR and AKT pathways, similarities exist between mTOR inhibitor and AKT inhibitors. However, AKT inhibitors are not yet approved for clinical use in the breast cancer setting and are currently in phase III trials [43].

The sole study investigating AKT inhibitors in our review demonstrated that two downstream proteins associated in the AKT pathway (pGSK3β and pPRAS40) were significantly downregulated. Consequently, there was once again a subsequent decrease in Ki67 expression denoting a reduction in the proliferative potential of the tumour. This is partly in contrast to preceding literature [44], whereby AKT inhibition did not produce a significant change in Ki67 expression, although it should be noted that in the study only two patients responded to therapy.

#### 4.1.6. Oestrogen

While estetrol (E4) did not elicit a significant change in Ki67 expression, there were some molecular indications of pharmacodynamic activity; namely a significant decrease in SHBG. As mentioned before, SHBG indicates the oestrogenicity of ER within the target tissue. As such, this would suggest that E4 had a partial ability to decrease the magnitude of oestrogenic effects within tumour tissue.

The significant downregulation of FSH following the full course of E4 therapy is further indication of this, given its role in stimulating oestrogen biosynthesis.

### 4.2. Strengths of the Present Review

This review was clearly focused on monotherapy. As a result, the pharmacodynamic biomarker transformations are more directly attributable to the drugs, which have induced such changes, and conclusions that are more meaningful can be drawn accordingly regarding these treatments in isolation than if we were to examine them in combination with other drugs. However, the authors do note that there are a number of newer therapies for ER+ breast cancer, which are only licensed for use in combination with another therapy.

Another area of strength compared to research was the focus upon the ER+ subset alone. By specifying the patient population, it provided the opportunity to more closely examine the therapeutic options available to this specialised group, whilst also addressing the most prevalent breast cancer subtype [45], further enhancing the clinical relevance of this review.

### 4.3. Limitations

Although overall, a large number of patients were included in this review, >80% of these came from study #15 which examined the biomarker Ki67; the number of patients in whom the other biomarkers were measured, is much smaller.

While the prerequisite of ER+ status was persistently applied, other characteristics of the specific populations of each trial varied. Some of the studies in our review [12,18,21] specified human epidermal growth factor receptor (HER) 2 negative status as a prerequisite for inclusion, whereas other studies did not. Furthermore, studies used various definitions of ER+ status. With future research and greater numbers of studies in the field, comparison at different levels of ER+ should be performed.

Another area of discrepancy within tumour classification is staging, with some studies only investigating early-stage [21], or stage I and II patients [11] while the rest made no such restriction. Given the established link between tumour staging and progressively worse response to neoadjuvant therapy [46], this is potentially relevant.

While there was sound reasoning in limiting the exposure to pharmacological therapy to a (mean) maximum of 31 days, in doing so it inevitably limited the literature search, possibly excluding studies, which may have been relevant. Likewise, the requirement for all participants to have been diagnosed with ER+ breast cancer does not allow for considerations of studies including ER+ patients within a larger cohort of breast cancer patients. This would have increased the total pool of patients available to include in this review.

The static nature of the dosages across all studies also prevented investigation of the effect of different de-escalation regimens upon biomarker expression, an area which has been studied in ER− HER2+ tumours in the WSG-ADAPT trial [47].

## 5. Conclusions

The adoption of WoO studies had been pervasive across a wide array of different drug classes, demonstrating a variety of mechanisms and targets used for the shared aim of inhibiting proliferation and combatting ER+ tumour growth.

The six classes of drug investigated in this review (AIs, SERMs, SERDs, mTOR inhibitors, AKT inhibitors, and oestrogens). There was a clear reduction in expression of ER, PR and Ki67 with endocrine therapies.

There is less evidence to support the impact of these drugs on the other biomarkers measured in this study: SHBG, pS6, CD34, pGSK3B, pPRAS40, pAKT, PTEN, caspase-3, BAX, BCL2, IGF1, FSH, prolactin.

Avenues for future work include investigation of biomarkers outside of ER, PR and Ki67 as well as consideration of combination therapy. The consideration of tumours demonstrating low ER expression (as opposed to ER+) also warrants further investigation, to determine if this tumour subtype behaves similarly in the WoO setting.

## Figures and Tables

**Figure 1 cancers-14-05027-f001:**
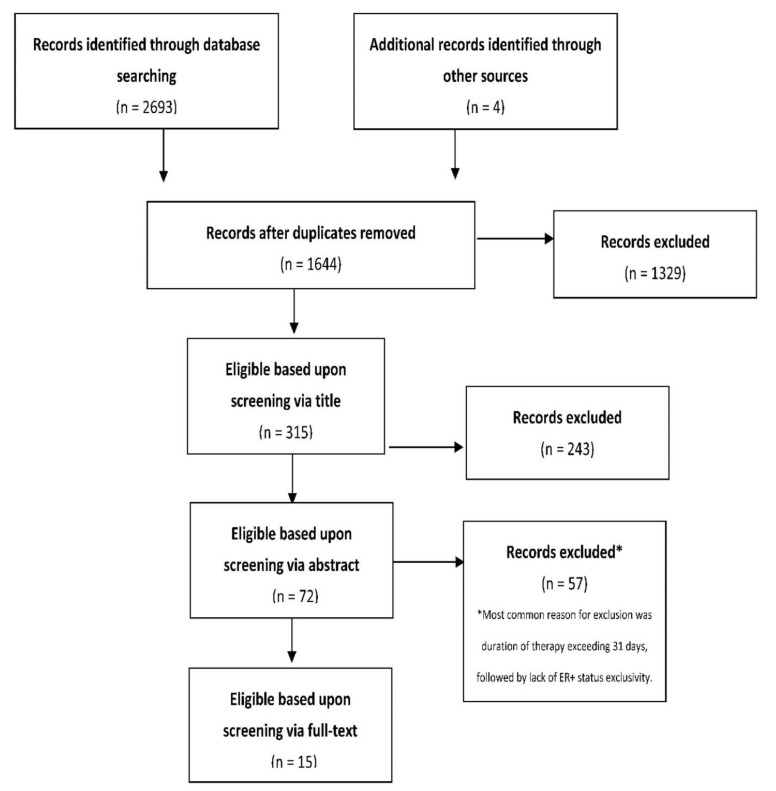
Flow diagram outlining the key stages in the study selection.

**Table 1 cancers-14-05027-t001:** Summary of the study characteristics of the 15 eligible WoO trials included in the systematic review. Treatments arms, which are italicised, were ineligible for inclusion.

Study No.	Author, Year	Tumour Characteristics	ER+ Status Threshold	Arm 1 Compound	Arm 2 Compound	Arm 3 Compound	Arm 4 Compound	Duration of Therapy (days)	Number of Patients	Biomarker Measured
1.	Arnaout et al., 2015 [7]	Post-menopausal, Stage II–III invasive carcinoma ≥ 2 cm	≥1% expression	Anastrozole	-	-	-	15–44	20	Caspase-3, Ki67
2.	Bedard et al., 2011 [10]	Post-menopausal	Allred Score 5–8	Letrozole	-	-	-	10–14	52	Ki67
3.	Cohen et al., 2017 [11]	Stage I–II invasive cancer	≥1% expression	Tamoxifen	-	-	-	7	44	Ki67
4.	Curigliano et al., 2016 [12]	Post-menopausal, Grade II–III HR+ HER2− invasive cancer	Not stated	Letrozole	*Letrozole + ribociclib*	*Letrozole + ribociclib*	-	14	14	Ki67
5.	Kisanga et al., 2004 [13]	None stated	≥20% expression	Tamoxifen (low dose)	Tamoxifen (medium dose)	Tamoxifen (high dose)	Placebo	28	120	Ki67, SHBG
6.	Loi et al., 2013 [14]	Post-menopausal cancer ≥ 2 cm	Not stated	Everolimus	*Letrozole*	*Letrozole + everolimus*	-	14	23	Ki67, pS6
7.	Mattar et al., 2011 [15]	None stated	Allred Score ≥2	Anastrozole	Tamoxifen	-	-	26	58	ER, PR, Ki67
8.	Morrogh et al., 2012 [16]	Post-menopausal breast cancer ≥ 1 cm	≥10% expression	Anastrozole	Placebo	-	-	10–11	26	CD34, Ki67
9.	Robertson et al., 2013 [17]	Post-menopausal primary	Not stated	Anastrozole	Fulvestrant	-	-	14–21	121	ER, PR, Ki67
10.	Robertson et al., 2020) [18]	Post-menopausal HER2− primary invasive cancer ≥ 1 cm	≥10% expression	AZD9496	Fulvestrant	-	-	5–14	46	ER, casPR, Ki67
11.	Robertson et al., 2020b [19]	Invasive breast carcinoma	Not stated	Capivasertib	Placebo	-	-	4.5	48	Ki67, pGSK3B, pPRAS40
12.	Sabine et al., 2010 [20]	Post-menopausal, early operable breast cancer	Not stated	Everolimus	-	-	-	11–14	32	Ki67, phospho-AKT
13.	Schmid et al., 2016 [21]	Post-menopausal HER2− invasive breast cancer	≥1% expression OR Allred Score ≥3	Anastrozole	*Anastrozole + pictilisib*	-	-	14	75	Caspase-3, Ki67, PR, PTEN
14.	Singer et al., 2014 [22]	Early-stage M_0_ breast cancer	Not stated	Estetrol (E4)	Placebo	-	-	14	30	Bax, Bcl-2, ERa, ERb, IGF-1, Ki67, PR, prolactin
15.	Smith et al., 2020 [23]	Post-menopausal operable primary, M_0_ breast cancer ≥ 1.5 cm	≥1% expression OR Allred Score ≥3	Anastrozole OR letrozole	Placebo	-	-	14	4480	Ki67

**Table 2 cancers-14-05027-t002:** Summary table of statistically significant (<0.05) biomarker changes associated with each class of drug given in WoO study. Raw data recorded where available to reflect the variability in assessing changes in expression.

Class of Drug(s)	Drugs	Molecular Biomarkers Significantly Changed from Baseline
AIs	Anastrozole, letrozole	CD34 ↑ (median 24% increase) ER ↓ (13% increase between mean expression) Ki67 ↓ (mean decreases of 29.1–80%) PR ↓ (37% increase between mean expression)
SERDs	AZD9496, fulvestrant	ER ↓ (mean decreases of 24.3–36.3%) Ki67 ↓ (mean decreases of 39.9–75.4%) PR ↓ (mean decreases of 33.3–68.7%)
SERMs	Tamoxifen	SHBG ↑ (Median increase of 18.6–25.4 ng/mL depending on smoking status) Ki67 ↓ (47.1% decrease between median expression, also mean decrease of 39.1% in Allred Scores)
mTOR inhibitors	Everolimus	Ki67 ↓ (mean decrease of 40.84%, 13/21 patients also had a “significant” [95% CI] decrease) pS6 ↓ (mean decrease of 76.6%)
AKT inhibitors	Capivasertib	pAKTc ↑ (721 patients had a “significant” [95% CI] decrease) Ki67 ↓ (9.6% decrease in number of Ki67-positive nuclei) pGSK3β ↓ (55.3 decrease in absolute H score)
Oestrogens	Estetrol (E4)	SHBG ↑ (*p* < 0.05, raw data not provided) ERβ ↓ (10/16 patients had a “significant” [*p* < 0.05] decrease) FSH ↓ (*p* < 0.05, raw data not provided) IGF-1 ↓ (*p* < 0.05, raw data not provided)

## Data Availability

Not applicable.

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
