# Peer review of "Molecular Biomarker Expression in Window of Opportunity Studies for Oestrogen Receptor Positive Breast Cancer—A Systematic Review of the Literature"

_cancers, 2022, doi:10.3390/cancers14205027_

Round 1

Reviewer 1 Report

General Comments:

The manuscript by James WM Francis, et al provided a landscape of molecular biomarkers in the window of opportunity treatment for ER+ breast cancer. The information from this manuscript would provide insightful perspectives into decision-making on neoadjuvant therapy for ER+ cancer, and would also suggest the molecular mechanism after the drug exposure. Therefore, this manuscript would be citable in current clinical and translational research in breast cancer patients.

The authors made a great attempt to explore the molecular biomarker repertoire in responding to various treatments in WoO studies. However, based on current clinical and pathological practice guidelines in breast cancer, there are several significant concerns needed to be addressed.

Major Comments:

1.      Inclusion and exclusion criteria on ER positivity: Please clarify the ER cutoff value to define the positive vs. negative.  

According to ASCO/CAP Guideline Update 2020 on Estrogen and Progesterone Receptor Testing in Breast Cancer. Cancer has a low level (1%-10%) of ER expression by IHC should report as ER-low expression, include comments: Cancer in this sample has a low level (1%-10%) of ER expression by IHC. There are limited data on the overall benefit of endocrine therapies for patients with these results, but they currently suggest possible benefit, so patients are considered eligible for endocrine treatment. There are data that suggest invasive cancers with these results are heterogeneous in both behavior and biology and often have gene expression profiles more similar to ER-negative cancers. 

2.      As the ASCO/CAP Guideline indicates: “for ER low expression cancer, there are limited data on the overall benefit of endocrine therapies for patients with these results”.

Please justify in the current manuscript whether the authors take this critical update into consideration and/or would be willing to design one arm of ER low expression cancer as a subset of ER+ cancer. The information on ER low expression cancer might provide greater insight into decision-making on endocrine therapy for this subset of ER+ cancer.

3.      For Table 1, under the category of “Patients”, it would be insightful to provide more information on their initial diagnosis such as histological types, tumor grade, et al.

4.  Table 2, under the category of “Molecular biomarkers significantly changed from baseline”. it would be more precise to add mathematical values (such as percentage value for tissue markers, and concentration value for circulating markers) to demonstrate the changes pre. vs. post-treatment.

5.      Besides the changes in molecular biomarkers, whether the authors also look for the “tissue pathological response” after the treatment. The attainment of pathological complete response (pCR) after neoadjuvant therapy has been taken as a surrogate marker for disease-free survival and overall survival. It would be informative if the authors were able to provide more information on histopathological evaluations such as pathological response.

Minor Comments:

1.      Page1, line 32. SHBG was mentioned twice. Please confirm. 

Author Response

General Comments:

The manuscript by James WM Francis, et al provided a landscape of molecular biomarkers in the window of opportunity treatment for ER+ breast cancer. The information from this manuscript would provide insightful perspectives into decision-making on neoadjuvant therapy for ER+ cancer, and would also suggest the molecular mechanism after the drug exposure. Therefore, this manuscript would be citable in current clinical and translational research in breast cancer patients.

The authors made a great attempt to explore the molecular biomarker repertoire in responding to various treatments in WoO studies. However, based on current clinical and pathological practice guidelines in breast cancer, there are several significant concerns needed to be addressed.

Thank you for your encouraging comments, which we have addressed as outlined below.

Major Comments:

  1. Inclusion and exclusion criteria on ER positivity: Please clarify the ER cutoff value to define the positive vs. negative.  

According to ASCO/CAP Guideline Update 2020 on Estrogen and Progesterone Receptor Testing in Breast Cancer. Cancer has a low level (1%-10%) of ER expression by IHC should report as ER-low expression, include comments: Cancer in this sample has a low level (1%-10%) of ER expression by IHC. There are limited data on the overall benefit of endocrine therapies for patients with these results, but they currently suggest possible benefit, so patients are considered eligible for endocrine treatment. There are data that suggest invasive cancers with these results are heterogeneous in both behavior and biology and often have gene expression profiles more similar to ER-negative cancers. 

We have collated data regarding the threshold for declaring ER positivity from each of the (15) studies included in the review and this has been added to Table 1. The ASCO/CAP guidelines have been included in the Discussion (page 4). 

  1. As the ASCO/CAP Guideline indicates: “for ER low expression cancer, there are limited data on the overall benefit of endocrine therapies for patients with these results”.

Please justify in the current manuscript whether the authors take this critical update into consideration and/or would be willing to design one arm of ER low expression cancer as a subset of ER+ cancer. The information on ER low expression cancer might provide greater insight into decision-making on endocrine therapy for this subset of ER+ cancer.

Due to the small number of studies included overall in this review, we have not formed a separate arm for ER low-expression tumours. We have highlighted this in the limitations (page 6).

  1. For Table 1, under the category of “Patients”, it would be insightful to provide more information on their initial diagnosis such as histological types, tumor grade, et al.

Tumour characteristics where published, have been collated and added to Table 1 as requested.

  1. Table 2, under the category of “Molecular biomarkers significantly changed from baseline”. it would be more precise to add mathematical values (such as percentage value for tissue markers, and concentration value for circulating markers) to demonstrate the changes pre. vs. post-treatment.

We have added the quantitative change in biomarker expression in Table 2 as requested. The disparity in methodology for assessing dynamic change in biomarker expression somewhat hindered direct comparison, however the precise method utilised has been stated each time, and this variety has been mentioned in the table legend.

  1. Besides the changes in molecular biomarkers, whether the authors also look for the “tissue pathological response” after the treatment. The attainment of pathological complete response (pCR) after neoadjuvant therapy has been taken as a surrogate marker for disease-free survival and overall survival. It would be informative if the authors were able to provide more information on histopathological evaluations such as pathological response.

      We have sought data pertaining to any measurement of tissue pathological response in the (15) studies however unfortunately none of them appeared to collect the necessary pCR data to make meaningful comment upon the histopathological impacts. This has been mentioned in the Discussion on page x.

Minor Comments:

  1. Page1, line 32. SHBG was mentioned twice. Please confirm. 

Amended.

Reviewer 2 Report

The review entitled 'Molecular biomarker expression in the window of opportunity studies for estrogen receptor-positive breast cancer- a systematic review of the literature aimed to study WoO trials that employed an array of drugs for Er+ breast cancer. The article is well structured and useful in the clinical context. I recommend this article for publication in its current form.

Author Response

Thank you for your enthusiasm for this article which we look forward to publishing with you.

Reviewer 3 Report

This is a review that aims to evaluate the role of biomarkers used as endpoints in window of opportunity trials in ER+ breast cancer patients treated with an endocrine-based treatment.

The authors found a total of 15 studies that matched the inclusion criteria they had established.

Following are some comments:

- In the Introduction: role and rationale for WOO should be better clarified, since these trials represent an excellent opportunity for better refining treatments to be investigated in Phase 3 clinical trials.

They should specify what is intended for WOO (e.g.: Short pre-surgical trials non-therapeutic studies in which patients are treated for 2 to 4 weeks….)

Their specific role (e.g.: they can be highly instructive in selecting or rejecting candidate approaches for phase III studies and defining the most appropriate patient populations…)

-  There are at least two other WOO that should be mentioned: CoopERA with the SERD giredestrant in postmenopausal women (this trial has a WOO phase with Ki67 and CCCA as biomarkers endopoints) and the WSG-ADAPT run-in phase (this trial utilized the Recurrence Score plus endocrine sensitivity testing to guide treatment. Endocrine sensitivity was determined by response of the proliferation marker Ki-67 to short-term preoperative endocrine therapy)

Author Response

This is a review that aims to evaluate the role of biomarkers used as endpoints in window of opportunity trials in ER+ breast cancer patients treated with an endocrine-based treatment.

The authors found a total of 15 studies that matched the inclusion criteria they had established.

Following are some comments:

Thank you for your comments.

- In the Introduction: role and rationale for WOO should be better clarified, since these trials represent an excellent opportunity for better refining treatments to be investigated in Phase 3 clinical trials.

The introduction has been amended to clarify the role of WoO trials in the context of subsequent drug development.

They should specify what is intended for WOO (e.g.: Short pre-surgical trials non-therapeutic studies in which patients are treated for 2 to 4 weeks….)

 We have re-enforced the non-therapeutic intentions of WoO trials (in comparison to neoadjuvant) and the comparatively shorter duration.

Their specific role (e.g.: they can be highly instructive in selecting or rejecting candidate approaches for phase III studies and defining the most appropriate patient populations…)

This comment is also now reflected in the introduction.

-  There are at least two other WOO that should be mentioned: CoopERA with the SERD giredestrant in postmenopausal women (this trial has a WOO phase with Ki67 and CCCA as biomarkers endopoints) and the WSG-ADAPT run-in phase (this trial utilized the Recurrence Score plus endocrine sensitivity testing to guide treatment. Endocrine sensitivity was determined by response of the proliferation marker Ki-67 to short-term preoperative endocrine therapy)

Thank you for bringing these studies to our attention. Both of these have now been included in the discussion for means of comparison (page 5 and 6).